# Commercial Gilthead Seabream (*Sparus aurata* L.) from the Mar Menor Coastal Lagoon as Hotspots of Microplastic Accumulation in the Digestive System

**DOI:** 10.3390/ijerph18136844

**Published:** 2021-06-25

**Authors:** Javier Bayo, Dolores Rojo, Pedro Martínez-Baños, Joaquín López-Castellanos, Sonia Olmos

**Affiliations:** 1Department of Chemical and Environmental Engineering, Technical University of Cartagena, Paseo Alfonso XIII 44, E-30203 Cartagena, Spain; drcampillo@gmail.com (D.R.); qlopezca@gmail.com (J.L.-C.); soniaespinar19@gmail.com (S.O.); 2C&C MedioAmbiente, E-30204 Cartagena, Spain; cycmedioambiente@cycmedioambiente.com

**Keywords:** microplastic, ingestion, fish, gilthead seabream, Mar Menor, marine pollution

## Abstract

This paper presents the results on the presence and characterization of microplastics (MP) in the gastrointestinal tract of gilthead seabream (*Sparus aurata* L.), a species of commercial interest from the Mar Menor coastal lagoon in Southeast Spain. This is the first time that microplastic ingestion is recorded in any species from this semi-enclosed bay. Stomach and intestine from a total of 17 specimens captured by local fishermen were processed, and microplastic particles and fibers found in all of them were displayed. Overall, 40.32% (279/692) of total isolated microparticles proved to be microplastics; i.e., <5 mm, as identified by FTIR spectroscopy. The average value by fish was 20.11 ± 2.94 MP kg^−1^, corresponding to average concentrations of 3912.06 ± 791.24 and 1562.17 ± 402.04 MP by kg stomach and intestine, respectively. Four MP forms were isolated: fiber (71.68%), fragment (21.15%), film (6.81%), and microbead (0.36%), with sizes ranging from 91 µm to 5 mm, an average of 0.83 ± 0.04 mm, and no statistically significant differences between mean sizes in stomach and intestine samples (*F*-test = 0.004; *p* = 0.936). Nine polymer types were detected, although most of fibers remained unidentified because of their small size, the presence of polymer additives, or closely adhered pollutants despite the oxidizing digestion carried out to eliminate organic matter. No significant correlation was found between main biological parameters and ingested microplastics, and high-density polyethylene (HDPE), low-density polyethylene (LDPE), polyethylene polypropylene (PEP), and polyvinyl (PV) were identified as the most abundant polymers. The average microplastic ingestion in this study area was higher than those reported in most studies within the Mediterranean Sea, and closely related to microplastic pollution in the surrounding area, although with a predominance of fiber form mainly due to fishery activities.

## 1. Introduction

The Mar Menor lagoon, located in the southeast of Spain, is a semi-enclosed hypersaline coastal system connected with the Mediterranean Sea through three shallow channels. It represents one of the environments of greater biological and socio-economic value in this region [1] and, as in other semi-enclosed bays, pollutants get accumulated to a greater extent than in open oceans [2]. The Mar Menor is also a very important habitat for many species of birds and fish, including several in danger of extinction, which coexist with increasing tourism, especially during the summer season along with management infrastructures that lag behind, bringing great pressure on the coastal lagoon [3].

After a previous manuscript reporting the abundance and distribution of microplastics in sand and sediments from urban, semi-urban, and natural beaches located at the Mar Menor [3], the aim of the present paper was to analyze the occurrence, type, amount, and characteristics of MP in the gastrointestinal tract of commercial gilthead seabream (CGS) (*Sparus aurata*) collected in this lagoon, one of the highest important commercial fish in the area [4]. In this sense, and to our knowledge, this paper represents the first study dealing with the interaction between microplastics and commercial fish in this important protected area. This was assessed by monitoring for microlitter (ML), defined as both plastic and non-plastic microparticles; i.e., small anthropogenic litter [5] and microplastics (MP), in the stomach and intestine of 17 individuals of CGS. The results allowed us to illustrate the link between human pressure and type of plastic pollution, the range of ingested amounts described in similar coastal waters, the relationship with MP concentration reported for sand and sediment in the Mar Menor, the difference in MP accumulation in stomach and intestine, and the main factors affecting the accumulation of different types of microplastics by CGS, providing a reference and baseline database for further studies such potential pathway for introducing both inorganic and organic pollutants into the food web [6,7,8], with potential health effects [9], in this zone.

## 2. Materials and Methods

### 2.1. Study Area and Sample Collection

Stomach and intestine of 17 individual of commercial gilthead seabreams (CGS) (*Sparus aurata* Linnaeus, 1758) were examined for microlitter (ML) such as cardboard, glass, bones, chitin shells, tissue remains, seeds or crystals of different salts, and microplastics (MP), both proved to be ingested by aquatic organisms [10]. They were all captured from the Mar Menor Sea, one of the largest hypersaline coastal lagoons in Europe, located in the Region of Murcia (Spain) and with a surface area of 135 km^2^ [3] (Appendix A). It is notified as a Special Protected Area (SPA) under de EU Birds Directive (79/409/EEC), the Ramsar Convention with the number 706 in October 1994, Site of Community Importance (SCI) under the EU Habitats Directive (92/43/EEC) and Special Protected Area of Mediterranean Interest (SPAMI) within the Barcelona Convention for the Protection of the Mediterranean Sea against Pollution since October 2001, among other local laws [3].

A total of 73 different fish species complete their life cycle in the Mar Menor, both entirely, temporarily, or by periodically visiting the lagoon [11]. *Sparus aurata*, with temperate fish larvae hatched from pelagic eggs, seasonally reaches the Mar Menor, using it as a nursery habitat and taking advantage of its high biological productivity [12]. Fish were caught by local fishermen mainly by gillnet fishing methods during three sampling campaigns: June–July 2018 (summer), November 2018–January 2019 (autumn–winter), and April–May 2019 (spring), as previously reported in other studies dealing with seasonal variation of pollutants in fish [13]. Ungutted CGS were stored in ice on the boat for their transportation and, once in the laboratory and prior to dissection, fish length (FL) (cm) and fish weight (FW) (g) were carried out to the nearest centimeter or milligram, respectively (Appendix A). In order to analyze how the environmental conditions of aquatic ecosystems affect fish populations Fulton’s factor was also calculated, according to the formula [14]: K = 100 (FW FL^−3^).

Gastrointestinal tract, including stomach and intestine, was dissected from fish body by means of a fine scalpel and stainless-steel tweezers (Appendix A). Stomach weight (SW) (g), and intestine weight (IW) (g) were recorded and stored in aluminum foil at −20 °C until further analyses. The whole dissection process was conducted in a clean fume hood to avoid airborne pollutants. Appendix A depicts FL, FW, SW, and IW for all analyzed specimens.

### 2.2. Microplastic Extraction

The content of both stomach and intestine, previously thawed at room temperature, was washed out with 100 mL of 120 g L^−1^ NaCl concentrated salt solution (2.05 M; *ρ* = 1.08 g mL^−1^) (Panreac, Barcelona, Spain), with the aid of a glass syringe. The mixture was placed into a 250 mL glass beaker and mechanically stirred in a jar-test device (300 rpm, 20 min). The following steps were similar to those reported for microplastic extraction in sand and sediments from the Mar Menor lagoon [3], except for organic matter digestion. Supernatant was vacuum filtered through a Büchner funnel using a paper filter (Prat Dumas, Couze-St-Front, France, diameter 110 mm, pore size 0.45 µm) (Appendix A). The funnel wall was twice washed with deionized water and filtered, and filters with isolated ML were placed into 120 mm glass Petri dishes with 10 mL 30% hydrogen peroxide (H_2_O_2_) solution at room temperature, to efficiently digest organic matter by orbital shaking (150 rpm, 10 min). Samples were dried overnight at 80 ºC in a forced air stove FD 23 (Binder GmbH, Tuttlingen, Germany), and kept in a desiccator with silica gel to avoid moisture until further analysis. All chemicals were of analytical reagent grade, and negative control samples or procedural blanks were carried out during the study, by filtering chemical reagents to determine any potential cross-pollution with airborne microplastics.

The risk of potential background contamination was also assessed. Only clothes made of natural fabric, clean cotton lab gowns and nitrile gloves were worn by the analysts during the whole process. The use of plastic lab devices was limited to the maximum, although it could not be entirely avoided. All tools and containers, including glassware and dissection apparatus, were thoroughly rinsed with tap water and three times with deionized water after their use, and covered with aluminum foil to prevent contamination [15]. All used containers were examined twice during the sampling campaign as procedural blanks, by vacuum filtering 100 mL deionized water to determine any potential microplastic contamination. No MP were isolated from blank samples.

### 2.3. Microplastic Observation and FTIR Analysis

Possible microplastic particle content collected from stomachs and intestines of CGS was examined under an Olympus SZ−61TR Zoom Trinocular Microscope (Olympus Co., Tokyo, Japan) coupled to a Leica MC190 HD digital camera and an image capturing software Leica Application Suite (LAS) 4.8.0 (Leica Microsystems Ltd., Heerbrugg, Switzerland), used for the analysis and recording of color, shape, and size of each particle in its longest dimension (Appendix A). Microparticles were visually classified as fiber (FB), fragment (FR), bead or pellet (BD), and sheet or film (FI). Besides, particulate microplastics, i.e., FR, BD, and FI, were named as MPP. Once the images were captured, particles were successfully isolated in 40-mm glass Petri dishes for further analysis by Fourier transform infrared spectroscopy (FTIR).

FTIR was used for the identification of functional groups and molecular composition of possible synthetic polymers in all identified microlitter. Microparticles were compressed in a diamond anvil compression cell, and spectra were acquired with a Thermo Nicolet 5700 Fourier transformed infrared spectrometer (Thermo Nicolet Analytical Instruments, Madison, WI, USA), provided with a deuterated triglycine sulfate, DTGS, detector and KBr optics (Appendix A). The spectra collected were an average of 20 scans with a resolution of 16 cm^−1^ in the range of 400–4000 cm^−1^. Spectra were controlled and evaluated by the OMNIC software without further manipulations, and polymers were identified by means of different reference polymer libraries. Microlitter particles matching reference spectra at a high level of certainty (>60%) were admitted as microplastics, and those with a lower level of agreement were subjected to further visual examination of spectra characteristics [16].

### 2.4. Statistical Analysis of Experimental Data

Statistical treatment of ML and MP data was carried out with SPSS (Statistic Package for Social Science) 26.0 statistic software (IBM Co. Ltd., Armonk, NY, USA). All data were expressed as mean ± standard error (SE). Pearson’s correlation coefficient (*r*) was used to test the extent to which values of two parameters were linearly correlated. The fitting performance of one-way analysis of the variance (ANOVA), between the abundance of MP and physical characteristics of the sampled fish, was computed by means of *F*-test. The least significance difference (LSD) test was applied when *F*-test reported rejection of null hypothesis (*H*_0_), in order to compare paired data and identify statistically significant differences. Critical value for statistical significance was set at 0.05 level.

## 3. Results and Discussion

### 3.1. Abundance and Morphology of Microplastics in CGS

A total of 692 ML were isolated from stomach and intestine of CGS, with average weight values of 2.15 ± 0.22 g and 9.63 ± 1.50 g, respectively, and average length and weight values of 32.24 ± 1.06 cm and 482.04 ± 38.37 g, respectively, as shown in Table 1. Figure 1a displays statistically significant differences between average concentration for stomach (10,495.11 ± 6572.53 ML kg^−1^) and for intestine samples (3542.52 ± 4202.44 ML kg^−1^) (*F*-test = 13.503, *p* = 0.001). ML average concentration per kg of analyzed fish was 50.47 ± 6.73 ML kg^−1^, with minimum and maximum values corresponding to 1.25 and 148.69 ML kg^−1^, respectively.

Every single isolated microparticle was analyzed by FTIR, and only 40.32% of this micropollutants; i.e., 279 items, proved to be MP, similar to that reported by [17] for sardines (47.2%), common pandoras (42.1%), and red mullets (32.0%) in Greek waters, by [18] in UK mussels (50%), or by [19] in the gastrointestinal tracts of 414 *Boops boops* (46.8%). Significant correlation between ML and MP concentrations were found for each individual (*r* = 0.763, *p* = 0.000).

Although most studies are based on visual identification, even with the naked eye, to separate microplastics from samples, it is the subsequent identification of polymers by spectroscopic techniques, e.g., FTIR or Raman, which allows us to successfully identify the type of polymer, avoiding overestimation due to false recognition of small fragments as plastics. It is this identification that helps us to understand the behavior and problems surrounding microplastic debris in the marine environment, such as transport capacity or location on stratification within the water column depending on the density of the polymer, or with its affinity for transferring pollutants through adsorption mechanisms [20]. However, due to the damage and potential for degradation during the microfiber digestion procedure, characteristics derived from [21,22] were also adapted for their identification, including no cellular or organic visible structures, color homogeneity properties and microfibers did not segment or fragmented when pressed.

Figure 2 depicts images of both ML and MP, in order to prove their likeness and the advantages of using a spectrometric technique in their differentiation.

All the analyzed specimens reported the presence of MP, both in the stomach and intestine, except for one individual without MP particles in the intestine. The available data on MP occurrence in the gastrointestinal tracts in different fish species show a wide range of results; i.e., [23] reported an occurrence of 44% (48 out of 110 specimens) either in the stomach or intestine of *S. aurata* from the Turkish territorial waters of the Mediterranean Sea, with a stereomicroscope and only 25 random particles analyzed by FTIR, and [24] a 23.3%, with visual identification and the hot needle technique. As previously reported, a standardization of MP extraction and analysis is required for further comparisons in this kind of studies [24].

The average concentration of MP accumulated in the digestive tract by kg of fish was 20.11 ± 2.94 MP kg^−1^, in the range reported by [25] in other carnivorous species from the Bohai Sea; i.e., *Scomberomorus niphonius* (10 ± 11 MP kg^−1^), *Sebastods schlegelii* (41 ± 47 MP kg^−1^), *Thryssa mystax* (87 ± 48 MP kg^−1^), *Eupleurogrammus muticus* (31 ± 18 MP kg^−1^), *Seriola aureovittata* (7 ± 6 MP kg^−1^), *Freeze myriaster* (20 ± 9 MP kg^−1^), *Lateolabrax maculatus* (17 ± 11 MP kg^−1^), *Paralichthys olivaceus* (10 ± 35 MP kg^−1^), or *Saurida elongate* (66 ± 35 MP kg^−1^). As depicted in Figure 3a, FB was by far the most abundant shape (71.68%; 15.04 ± 2.44 FB kg^−1^ fish), followed by fragment (FR) (21.15%; 3.50 ± 1.02 FR kg^−1^ fish), film (FI) (6.81%; 1.52 ± 0.50 FI kg^−1^ fish), and microbead (BD) (0.36%; 0.04 ± 0.04 BD kg^−1^ fish). On the contrary, beaches surrounding the Mar Menor coastal lagoon were mainly polluted with fragmented forms, with a higher average concentration in urban (33.5 ± 9.4 items kg^−1^ dry sediment) than in natural and semi-natural beaches (27.8 ± 7.3 items kg^−1^ dry sediment) [3]. A lack of environmental education on fisherman results in derelict fishing nets, lines, and ropes in the coastal lagoon, that continue to break down and entering the food web. In fact, all these items represent a considerable proportion of marine litter worldwide (18.5%) [26], and in fish collected from the Mediterranean coast, as reported by [27] (71%); [23] (70%); [28] (83%); or [19] (81.95%). According to [29], feeding habits should be also taken into consideration, with a higher amount of fibers ingested by omnivorous fish than herbivores and carnivores. *Sparus aurata* has a widely varied diet, being omnivorous at a juvenile stage and with a predatory feeding behavior on a further stage [30].

The average number of microplastics in stomachs and intestines of CGS ranged between 0 and 34 items, totally agreeing with the results reported by [31] also in gilthead seabream after 45 days’ exposure to virgin microplastics, with an average number of microplastic particles in the fish intestines and stomachs ranging between 0 and 34 for all plastic types analyzed. Average numbers of MP by stomach and intestine were 7.88 ± 1.77 and 8.53 ± 2.19, respectively, without statistically significant differences (*F*-test = 3.133, *p* = 0.053), being higher than those reported by [23] in 110 specimens of *Sparus aurata* from the Turkish coast, with average values of 1.53 and 1.47 MP for stomach and intestine, respectively. Average abundance of 16.41 MP per fish was higher than 1.2 MP per individual reported by [32] in 761 individuals from mesopelagic families; 2.3 MP per individual in 64 Japanese anchovies, with a maximum of 15 pieces per individual [33]; 1.2 MP per individual in 189 fish specimens from the Amazon River estuary [34]; 2.14 MP per individual reported by [25] in the Bohai Sea; or 4.3 items per wild mullet reported by [15]. The authors of [35] reported 88 particles in the whole digestive tract from three adult female whales stranded on the North and West Ireland coast, and [36] reported a concentration of 19.7 microplastics per individual, in gut contents of 11 species from an Argentinean coastline estuary. Microplastic ingestion value identified for *Sparus aurata* in this study was also higher than those reported in the Mediterranean basin; i.e., 3.75 items per fish in *Boops boops* from the Balearic Islands [37]; 0.34 items per fish in 125 individuals of blackmouth catshark [38]; 0.88 and 0.20 items per individual in the Spanish peninsular coast and the Balearic Islands, respectively [39]; or 2.51 ± 0.02 MP per individual in *B. boops* gastrointestinal tract [19]. It seems that microplastics in CGS from Mar Menor are in higher concentration than most reported studies, with a frequency of occurrence of 100% in examined specimens. The absence of a standardized approach for sample collection, extraction method, and analysis makes comparability between studies a difficult task. Moreover, the results reported in many studies do not clarify whether they refer to the wet or dry weight of the whole sample, or to the wet or dry weight of the tissue or organ containing MP. Jovanović, 2017 [40] also reported different percentages of ingested MP by fish according to the considered length of particles, dropping from 23% down to 2.6% if fibers were excluded from counting. Even with all these considerations, our findings reflect a microplastic ingestion closely related to microplastic pollution in the surrounding beaches of the semi-enclosed coastal lagoon, with limited self-cleaning abilities. An important runoff conveys from a terrestrial source, with an intensive agriculture land-use and terrestrial application of sewage sludge as a fertilizer [3].

Calculated average microplastic concentrations for stomach (3912.06 ± 791.24 MP kg^−1^) and intestine (1562.17 ± 402.04 MP kg^−1^) showed statistically significant differences (*F*-test = 7.010, *p* = 0.012), as presented in Figure 1a, with a statistically significant increase in the mean ratio FB:MP from stomach (0.63 ± 0.08) to intestine (0.85 ± 0.06) (*F*-test = 4.551, *p* = 0.041). Therefore, although there is a statistically significant lower concentration of MP in intestine samples than in the stomach ones, the proportion of fibers is higher in intestine than in stomach, which means a selection in the passage of these forms to the detriment of particulate forms. Huang, 2020 [41] described a higher percentage of MP in intestine (47%) than in stomach (30%) in fish from a Chinese mangrove wetland, referring fibers to be more abundant than fragments. As reported by [40], it is hard to say the daily microplastic ingestion load of a fish in its natural environment, as such studies do not exist.

Individuals collected during the spring period proved to have statistically significant length and weight average values higher than those collected during the other sampling campaigns, as depicted in Appendix A. Conversely, specimens collected during that period displayed the lowest average ML (13.53 ± 3.49 ML kg^−1^) and MP (4.57 ± 0.84 MP kg^−1^) concentrations, (*F*-test = 15.511, *p* = 0.000 for ML; and *F*-test = 13.317, *p* = 0.000 for MP), with also statistically significant differences between spring and autumn–winter, and spring and summer (*LSD tests*, *p* < 0.05), as presented in Figure 1b. Statistically significant differences were also evident for FB (*F*-test = 8.465, *p* = 0.001) and MPP (*F*-test = 4.513, *p* = 0.019) by season, although higher average values were reported for MPP (9.40 ± 3.10 MPP kg^−1^) during summertime, and for FB (22.67 ± 4.86 FB kg^−1^) during the autumn–winter period (Figure 1b). These differences should be explained by human activities and land use.

Fibers might originate from broken fishing lines or nets because of fishery activities in the Mar Menor, taking place all over the year; meanwhile, the increased content of particulate microplastics should be related to a touristic and intensive recreational use of the coastal lagoon during the summer season, when big plastics are also exposed to extreme degradation conditions because a light-induced oxidation, the main responsible process for plastic fragmentation in the marine environment [42,43]. Although there are other factors that might help us understand these results, as suggested by other authors who have also related the increment of population and weather conditions during the hot period with the occurrence of different pollutants in the aquatic compartment [13] or the influence that changes in salinity and temperature could have on the distribution of MP within the water column [44]. Tsangaris, 2020 [19] also reported a significant and positive relationship between ingested microplastics and the degree of anthropization. Considering all these results, we can verify that most of microplastics detected in this study were secondary microplastics, from the continuous fragmentation of oversized plastics.

No statistically significant dependence was highlighted between the main fish biological parameters, i.e., fish length, fish weight, stomach weight, intestine weight, or K and the number of ingested ML or MP (Pearson’s *r*, *p* > 0.05). Similar results were reported by [17] in marine organisms from the Northern Ionian Sea, [24] in edible fish species collected in the Mediterranean Sea, or [19] in Osteichthyes also in the Mediterranean Sea.

### 3.2. Size, Color, and Polymer Distribution

The size of MP ranged from 91 µm to 5 mm, both values corresponding to FB isolated in the intestine, with an average length of 0.83 ± 0.04 mm, longer than that reported in other fish studies carried out by [24] (0.10 mm), or [45] (0.149 mm), and shorter than that reported by [15] in wild and captive mullets (1.18 mm). MP average length was shorter than in our previous study in sand and sediments of Mar Menor [3], with a mean value of 1.4 ± 0.1 mm. There was no statistically significant difference between mean sizes of MP in stomach (0.84 ± 0.06 mm) and intestine samples (0.83 ± 0.06 mm) (*F*-test = 0.004; *p* = 0.936) (Table 1). Figure 3b depicts the distribution of size categories according to stomach and intestine samples. The most common microplastic size class in the stomach was 200–400 µm (31.34%), and 1–2 mm (26.21%) for intestine, being 68.10% of all isolated microplastics under 1mm in size. No correlations were found between microplastic size and fish length, fish weight, stomach weight, intestine weight, or K (Pearson’s *r*, *p* > 0.05).

The most common color for microplastic particles was white (63.08%), both for stomach (67.16%) and intestine samples (59.31%) (Figure 3c), representing more than half of total microplastics isolated, and followed by blue (23.30%), red (5.38%), black (3.23%), and brown (2.87%). Blue fibers were mainly broken fishing lines (Figure 2i,k). Only a few microplastic particles proved to be yellow (1.43%), green (0.36%), and pink (0.35%). The ability of predators for ingesting MP with colors resembling their prey has been thoroughly described [46], exhibiting different inhibition effects according to the color [47]. As reported by [48], white color could be preferentially consumed by fish, instead of black or red particles, due to directly mistaking microplastics to natural prey in size and appearance. In fact, as presented in Appendix A, transparent MP were advantageously isolated in early juveniles of CGS, possibly due to a result of similar appearance with transparent shrimp and prawns. Gilthead seabream has a widely varied diet [30], with a preference for live prey [49], and the uptake of microplastics can occur indirectly via the consumption of shellfishes, mollusks, annelids, copepods, rotifers, or other fishes containing these emerging pollutants. In any case, microplastic color could be affected by the digestion procedure, being reported that yellow and brown coloration observed in microplastic samples from wastewater due to organic matter had disappeared after H_2_O_2_ digestion [50].

Nine different polymer types were detected across all CGS samples, as presented in Figure 3d, although only four of them were isolated in intestine samples, i.e., low-density polyethylene (44.12%), high-density polyethylene (32.35%), polyethylene propylene (14.71%), and polyvinyl (8.82%). Similar results were reported by [15], with nine polymer types identified in mullet samples. Figure 4 shows some examples of distinctive absorption bands for FTIR spectra, both for microplastic samples and standard polymer from reference libraries. Polyethylene and polypropylene have proved to be among the most demanded polymer types by plastic converters all over the world [51], and commonly reported in the digestive tract of fish [14,15,17,19,52]. Stomach samples retained a wider variety of polymers than intestine samples, including low-density polyethylene (47.83%), high-density polyethylene (26.09%), polyethylene propylene (6.52%), and polyvinyl (6.52%), polyester (4.35%), acrylate (2.17%), polyurethane (2.17%), polystyrene (2.17%), and polytetrafluoroethylene (2.17%). Polyvinyl and propylene fibers may be originated from fishing lines and ropes used in ships [53].

In addition, polyvinyl ester resins are increasingly used as matrix materials, especially for marine composites, because of superior mechanical properties, corrosion resistance and low cost [54]. As previously indicated, and in an effort to positively recognize the detected MP by FTIR, the polymer constitution of 71.32% of FB remained unidentified; i.e., 65.67% for stomach, and 76.55% for intestine samples. The reasons masking the signal and hindering polymeric identification by spectrometric analyses have been widely reported, mainly due to small size, the presence of polymer additives [55], pigments [56], dyes [57], or closely adhered pollutants and remaining organic matter [58]. Moreover, the digestion procedure undergone to safety eliminate organic matter may also alter the distinct peaks in the FTIR spectrum, hindering comparisons with reference libraries [57]. Digestion with oxidizing hydrogen peroxide (H_2_O_2_) has been reported to have limited damages to integrity of the main polymer types, although it may affect some of their physical properties such as transparency and thickness [59], also resulting in incomplete soft tissue digestion that may interfere with MP identification [60]. Difficulties in fiber classification have been previously reported by [61] (62.3%) in airborne samples, or by [62] (63.9%) in wastewater samples. Karami, 2017 [6] reported 13.1% of unidentified particles (8 out of 16) in commonly consumed dried fish species, and [45] reported 17.8% of unidentified particles (10 out of 56) in commercial marine fish from Malaysia, both after characterization by Raman and energy-dispersive X-ray spectroscopy.

Absorption bands from LDPE (Figure 4a) include an asymmetric vibration of CH_2_ groups between 2930–2850 cm^−1^ wavenumbers, bending C-C bond between 1450–1470 cm^−1^, and rocking in plane for CH_2_ groups at 700–750 cm^−1^. The prominent peak in microplastic sample spectrum between 1710–1720 cm^−1^ should be related to the carbonyl group stretching; as a result of the oxidative treatment using H_2_O_2_, that may introduce a carbonyl functional group within the alkane structure [63]. Besides, LDPE has proven to undergo weather-induced degradation, being the formation of carbonyl group the most obvious chemical induced effect [64].

The FTIR spectrum for HDPE is shown in Figure 4b, with its characteristic absorption bands at 2900–2950 cm^−1^ and 2800–2850 cm^−1^ wavenumbers, corresponding to CH_2_ group asymmetric and symmetric stretching, respectively, 1550–1600 cm^−1^ and 1450–1500 cm^−1^ for bending deformation, 1350–1400 cm^−1^ for wagging deformation, and 700–750 cm^−1^ due to rocking deformation [65]. As depicted in Figure 4b, there is no new absorption band in the microplastic sample, except an increase in the intensity of the absorption band at 700–750 cm^−1^ wavenumbers [66], and a negative evolution in wagging deformation because of crosslinks between polymeric chains [67]. FTIR spectrum for PEP shares many characteristics previously reported for both LDPE and HDPE spectrum.

## 4. Conclusions

This research studied the presence, abundance, morphology, and composition of microplastics collected in stomach and intestine of commercial gilthead seabream (*Sparus aurata*) collected from the Mar Menor coastal lagoon. We can state that microplastics were detected in all analyzed individuals, with an average concentration of 20.11 ± 2.94 MP kg^−1^, similar to that reported in other semi-enclosed seas and lower than that reported for the same species in open seas. Microplastic concentration proved to be higher for stomach (3912.06 MP kg^−1^) than for intestine samples (1562.17 MP kg^−1^), and ingested microplastics consisted primarily of fiber (71.68%), followed by fragment (21.15%), film (6.81%), and microbead (0.36%). The highest average value for fiber form in autumn-winter fish samples (22.67 FB kg^−1^) could be originated from fishery and domestic wastewater, among other human activities. Conversely, particulate microplastics were mainly isolated in fish collected during the summer (9.40 MPP kg^−1^), coinciding with massive tourism in the coastal zone and a greater fragmentation of plastic waste due to weather conditions. High-density polyethylene (HDPE), low-density polyethylene (LDPE), polyethylene polypropylene (PEP), and polyvinyl (PV) were identified as the most abundant polymers ingested by commercial gilthead seabream, with a variety of colors that demonstrate their multiple origin. Taking into account the important role of fish in human nutrition, especially in coastal areas, further research on microplastic monitoring is needed and advisable, providing useful information for transfer potential assessment through the aquatic food web, both from microplastics themselves and from hazardous chemical as that they may transport, and establishing the risk for human exposure from commercially important species.

## Figures and Tables

**Figure 1 ijerph-18-06844-f001:**
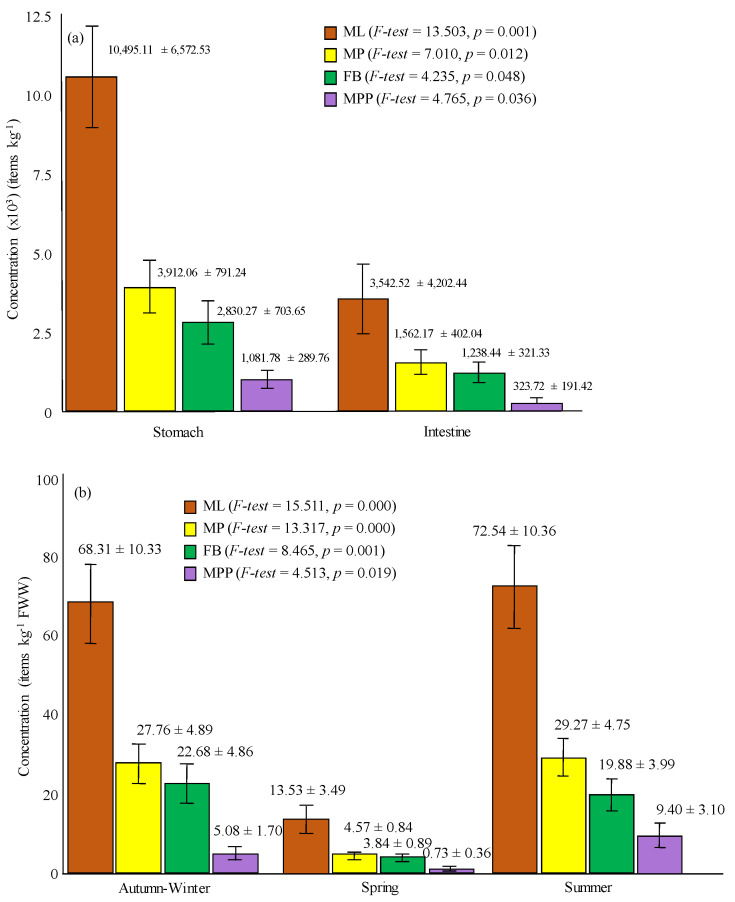
Average concentrations of: (**a**) microlitter (ML), microplastic (MP), fiber (FB), and particulate microplastic (MPP) in fish stomachs and intestines; (**b**) ML, MP, FB, and MPP according seasons (error bars represent standard error).

**Figure 2 ijerph-18-06844-f002:**
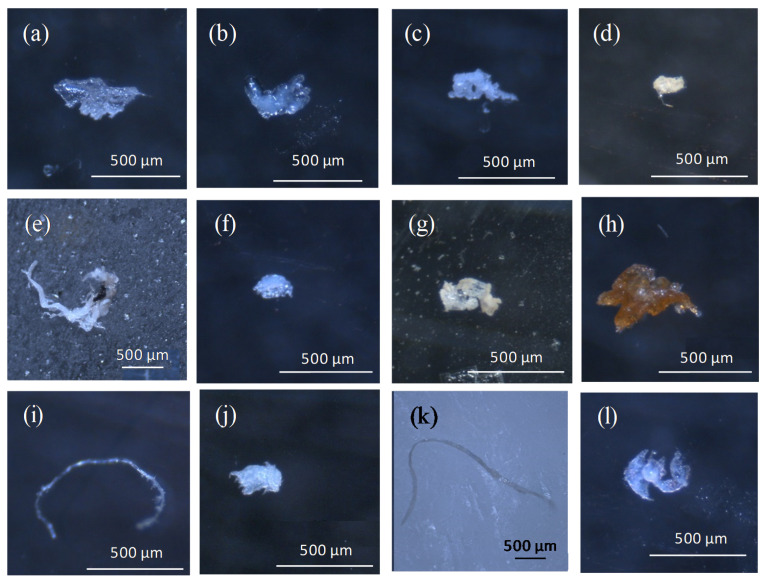
Microlitter (**a**–**d**) and microplastics (**e**–l) in stomach (S) and intestine (I) of commercial gilthead seabream collected in the Mar Menor: (**a**) carboxymethylcellulose, sodium salt (I); (**b**) glycerol monooleate (S); (**c**) sodium stearate (S); (**d**) paraffin wax (I); (**e**) poly(ethylene) (low density) (S); (**f**) eva foam concentrate,10% azodicarbonamide (S); (**g**) polyethylene, linear (I); (**h**) poly(ethylene) (high density) (S); (**i**) unknown fiber (I); (**j**) polyethylene type F (S); (**k**) poly(ethylene) (low density) (S); (**l**) poly(ethylene:vinyl acetate) (9% vinyl acetate) (S).

**Figure 3 ijerph-18-06844-f003:**
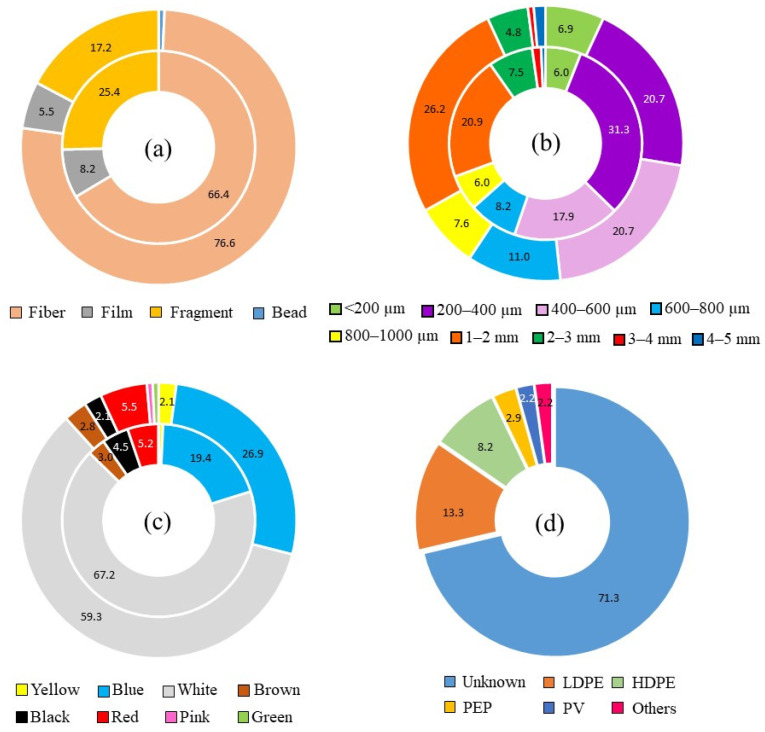
Accumulated percentages of MP ingested by CGS: (**a**) shapes; (**b**) size categories, based on Spanish Environmental Ministry classification; (**c**) colors; (**d**) polymer types (inner ring stands for stomach samples, and outer ring stands for intestine samples).

**Figure 4 ijerph-18-06844-f004:**
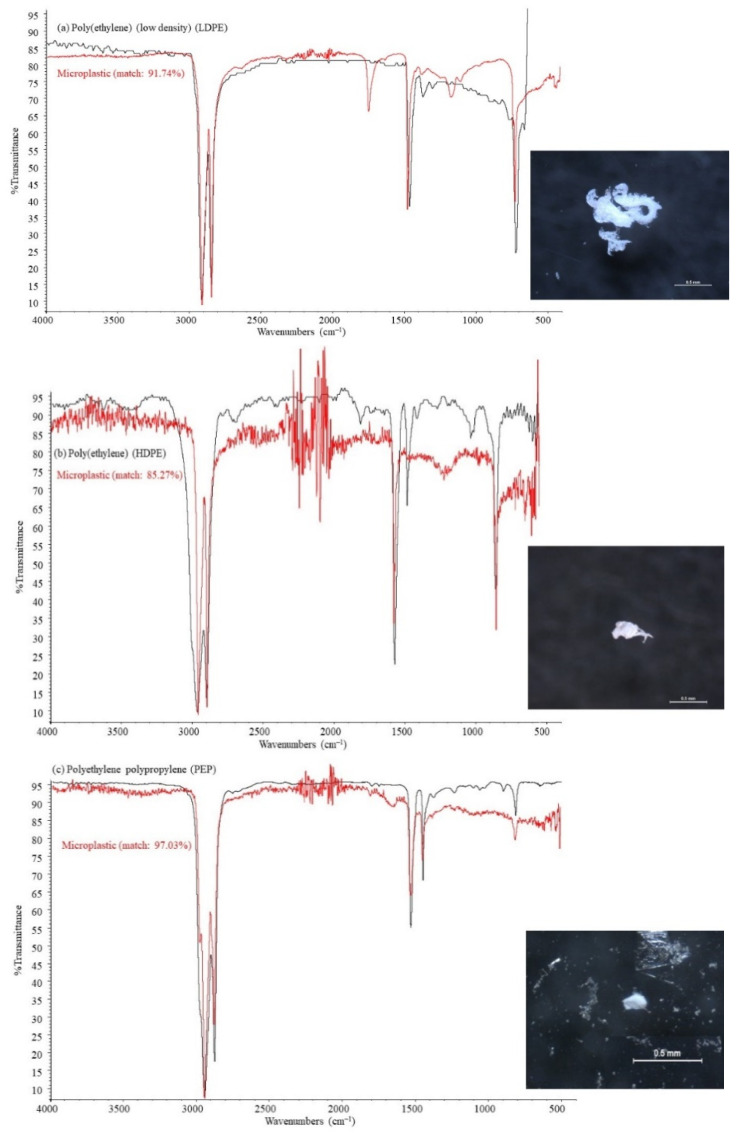
FTIR standard spectra (black lines) and microplastic samples (red lines) for: (**a**) low-density polyethylene (LDPE); (**b**) high-density polyethylene (HDPE); and (**c**) polyethylene polypropylene (PEP).

**Table 1 ijerph-18-06844-t001:** Average values (± standard error) of diferent parameters analysed in commercial gilthead seabream (CGS). (ML: microlitter, MP: microplastics, FB: fibers; MPP: particulate microplastics).

Paramenters Analysed	
Number of individuals examined	17
Fish length (cm)	32.24 ± 1.06
Fish weight (g)	482.04 ± 38.37
Stomach weight (g)	2.15 ± 0.22
Intestine weight (g)	9.63 ± 1.50
Digestive tract (g)	11.44 ± 1.77
Fulton’s condition factor (K)	1.37 ± 0.01
Number of individuals containing ML	17
Number of individuals containing MP	16
ML number	692
ML average concentration (items kg^−1^)	
(a) ML in stomach	10,495.11 ± 1594.07
(b) ML in intestine	3542.52 ± 1019.24
(c) ML in digestive tract	5034.59 ± 1067.50
MP number	279
MP average size (µm)	
(a) Stomach	0.84 ± 0.06
(b) Intestine	0.83 ± 0.06
(c) Digestive tract	0.83 ± 0.04
MP average concentration (items kg^−1^)	
(a) MP in stomach	3912.06 ± 791.24
(b) MP in intestine	1562.17 ± 402.04
(c) MP in digestive tract	2010.71 ± 414.64
(d) Fibers (FB) in stomach	2830.27 ± 703.65
(e) Fibers (FB) in intestine	1238.44 ± 321.33
(f) Fibers (FB) in digestive tract	1538.36 ± 354.98
(g) Particulate microplastics (MPP) in stomach	1081.78 ± 289.76
(h) Particulate microplastics (MPP) in intestine	323.72 ± 191.42
(i) Particulate microplastics (MPP) in digestive tract	472.35 ± 177.26

## Data Availability

The datasets generated and analyzed during the current study are not publicly available, but they are available from the corresponding author on reasonable request.

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
