# Peer review of "Commercial Gilthead Seabream (Sparus aurata L.) from the Mar Menor Coastal Lagoon as Hotspots of Microplastic Accumulation in the Digestive System"

_ijerph, 2021, doi:10.3390/ijerph18136844_

Round 1
Reviewer 1 Report
The manuscript on the overall is rather interesting and adds useful data for the comprehension of the fate of MPs in marine organisms of important commercial value and for an area as the Mar Menor coastal lagoon of environmental value.
The paper presents the results on the presence and characterization 13 of microplastics (MP) in the gastrointestinal tract of gilthead seabream (Sparus 14 aurata L.), a specie of commercial interest from the Mar Menor coastal lagoon 15 in southeast Spain. This is the first time that microplastic ingestion is recorded 16 in any specie from this semi-enclosed bay. Stomach and intestine specimens captured by local fishermen were assayed for microplastic particles and fibres presence.
The main strengths of the paper are the originality of the research since for the first time are provided data on microplastics in the gilthead seabream of the coastal lagoon of Mar menor in southeast Spain.
The main weakness is the first part of the introduction where there is an initial part on the occurrence of microplastics in sea environment that is rather repetitive and commonly found in hundreds of similar papers.
Minor recommendations:
Lines 43 to 73: delete since they are obvious and too often reported in the literature, start with the line 74.
Lines 246 to 261: rephrase since it is confusing and fuzzy. I suggest the authors to rewrite them and to clearly indicate what they want to state even when they cite references 36 and 37. It is difficult to understand the characteristics derived from these citations.
I do not like to report a direct citation to an author indicating just strictly the sequential number of the citation. I suggest indicating the name of the Author or authors and then in square parenthesis the number.
Author Response
Response to Reviewer 1 Comments
Point 1: Lines 43 to 73: delete since they are obvious and too often reported in the literature, start with the line 74.
Response 1: Lines 43 to 73 have been deleted, and references have been modified and renumbered
Point 2: Lines 246 to 261: rephrase since it is confusing and fuzzy. I suggest the authors to rewrite them and to clearly indicate what they want to state even when they cite references 36 and 37. It is difficult to understand the characteristics derived from these citations.
Response 2: Paragraph has been rewritten and the characteristics derived from quotations 36 and 37 are listed below the quotation (lines 224-239). Find below the new paragraph:
“Although the vast majority of studies are based on visual identification, even with the naked eye, to separate microplastics from samples, it is the subsequent identification of polymers by spectroscopic techniques, e.g. FTIR or Raman, which allows us to successfully identify the type of polymer, avoiding overestimation due to false recognition of small fragments as plastics. It is this identification that helps us to understand the behavior and problems surrounding microplastic debris in the marine environment, such as transport capacity or location on stratification within the water column depending on the density of the polymer, or with its affinity for transferring pollutants through adsorption mechanisms [20]. However, due to the damage and potential for degradation during the microfiber digestion procedure, characteristics derived from [21] and [22] were also adapted for their identification, including no cellular or organic visible structures, color homogeneity properties and microfibers did not segment or fragmented when pressed.”
.
Point 3: I do not like to report a direct citation to an author indicating just strictly the sequential number of the citation. I suggest indicating the name of the Author or authors and then in square parenthesis the number.
Response 3: According to the publication standards of the journal, references to authors should be made with the sequential number of the citation.
Reviewer 2 Report
Review
Journal: Int. J. Environ. Res. Public Health
Manuscript Title: Commercial gilthead seabream (Sparus aurata L.) from the Mar Menor coastal lagoon as hotspots of microplastic accumulation in the digestive system
The manuscript deals with the presence and characterization of microplastics (MP) in the gastrointestinal tract of gilthead seabream (Sparus aurata L.), a species of commercial interest, collected from the Mar Menor coastal lagoon in southeast Spain.
Authors underline that there isn’t information on plastics in species from this lagoon.
The paper is worthy of publication
Only minor suggestions:
Line 70: Authors should cite the following articles which examined Sparus aurata and coastal fauna samples.
- Gugliandolo, E.; Licata, P.; Crupi, R.; Albergamo, A.; Jebara, A.; Lo Turco, V.; Giorgia Potortì, A.G.; Mansour, H.B.; Cuzzocrea, S.; Di Bella, G. Plasticizers from microplastics in Tunisian marine environment. Front. Mar. Sci. 2020, 7, 928.
- Lo Brutto, S.; Iaciofano, D.; Lo Turco, V.; Potortì, A.G.; Rando, R.; Arizza, V.; Di Stefano, V. First Assessment of Plasticizers in Marine Coastal Litter-Feeder Fauna in the Mediterranean Sea. Toxics 2021, 9, 31.
- Line 107: Change in: Stomach and intestine.. both singular
- Line 108: Authors should specify which type of microlitter they examined. Pay attention, litter commonly includes detritus
- Line 123 At the beginning of a sentence you should use the not contract form of scientific name, thus no aurata, but Sparus aurata
- Line 170: Possible instead of Posible
- Line 238: Every single isolated microparticle was analyzed by FTIR, and only 40.32% of this micropollutants…. Maybe “to be” lacks
Line 383 …… These differences should be explained by human activities and land use…. Maybe other factors can influence the variability of the ingested MPP along seasons. I would improve such discussion.
Final question: Do the S.aurata samples come from aquaculture farms? If they do, it should be specified. Are farms close the sampling site?
Author Response
Answers to Reviewer #2:
Comments on ijerph-1259620
Point 1: Line 70: Authors should cite the following articles which examined Sparus aurata and coastal fauna samples.
- Gugliandolo, E.; Licata, P.; Crupi, R.; Albergamo, A.; Jebara, A.; Lo Turco, V.; Giorgia Potortì, A.G.; Mansour, H.B.; Cuzzocrea, S.; Di Bella, G. Plasticizers from microplastics in Tunisian marine environment. Front. Mar. Sci. 2020, 7, 928. doi.org/10.3389/fmars.2020.589398
- Lo Brutto, S.; Iaciofano, D.; Lo Turco, V.; Potortì, A.G.; Rando, R.; Arizza, V.; Di Stefano, V. First Assessment of Plasticizers in Marine Coastal Litter-Feeder Fauna in the Mediterranean Sea. Toxics 2021, 9, 31.doi: 10.3390/toxics9020031.
Response 1: The references have been included. Line 75. Reference [7] y [8]. Find below the new paragraph:
“(…)such potential pathway for introducing both inorganic and organic pollutants into the food web [6, 7, 8], with potential health effects [9], in this and similar areas.”
Point 2: Line 107: Change in: Stomach and intestine.. both singular
Response 2: “S” has been removed from intestines. Line 79
Point 3: Line 108: Authors should specify which type of microlitter they examined. Pay attention, litter commonly includes detritus
Response 3: A short description about microlitter has been included (lines 81-82). Find below the new paragraph:
“microlitter (ML) such as cardboard, glass, bones, chitin shells, tissue remains, seeds or crystals of different salts, etc.,”
Point 4: Line 123 At the beginning of a sentence you should use the not contract form of scientific name, thus no aurata, but Sparus aurata
Response 4: S.aurata has been corrected for Sparus aurata (line 94)
Point 5: Line 170: Possible instead of Posible
Response 5: Posible has been changed for “possible” (line 155)
Point 6: Line 238: Every single isolated microparticle was analyzed by FTIR, and only 40.32% of this micropollutants…. Maybe “to be” lacks
Response 6: All ML isolated after visual inspection by trinocular microscopy were analyzed with FTIR in order to identify whether the analyzed material was plastic or not, and following the standard criteria published by Frias et al. (2016), i.e. considering as MP the spectra with a percentage of coincidence higher than 70% between the analyzed sample and the reference spectrum, examining individually the cases that were below this percentage and discarding samples where the peaks did not correspond to identifiable material or below 30% coincidence. In our study, of the total of 692 ML isolated, 279 were found to be MP (40.32%).
Point 7: Line 383 …… These differences should be explained by human activities and land use…. Maybe other factors can influence the variability of the ingested MPP along seasons. I would improve such discussion.
Response 7: Other aspects have been slightly considered. Lines 374-380. Find below the new paragraph:
Although there are other factors that might help us understand these results, as suggested by other authors who have also related the increment of population and weather conditions during the hot period with the occurrence of different pollutants in the aquatic compartment [13] or the influence that changes in salinity and temperature could have on the distribution of MP within the water column [44].
Point 8: Final question: Do the S.aurata samples come from aquaculture farms? If they do, it should be specified. Are farms close the sampling site?
Response 8: No. The fishes were caught by local fishermen mainly by gillnet fishing methods. (Line 97)
Reviewer 3 Report
This study details the presence of microplastic and microlitter in the digestive tract of Sparus aurata in an enclosed coastal system with microplastic present in sand and sediments – Mar Menor (Múrcia, Spain). This species has high economic value because is the main fish consumed in the area by humans with high impacts into local trophic chain also.
M& M
-17 specimens are a very small sample of fish and being all from the same fisherman (same site) is not the most random sample. I also know that the authors intent to contain the study to the Mar Menor area.
-It is not also clear, that the all sample is 17 organisms or 17 were collected in each season? Please clarify it in the text.
- Please add some information about Fulton’s factor and why it is used?
-Ln 178 - typing error “Posible” – did the author mean possible?
Ln 231 – Please add the abbreviation used in the legend of the Table 1.
Ln 254-261- This sentence needs to be better explained because it is not clear to the reader.
Ln 282 and after in the manuscript when the author mentioned MP quantification include or not ML? I think that need to be clear to the reader otherwise what’s the purpose of differentiating it the M& M section?
Ln 344 . This sentence need English spell check.
Ln 397-399. What means being secondary microplastics. The authors did not mention it before not even in Introduction section. That needs to be clarified and its implications of being secondary…
Author Response
Answers to Reviewer #3:
Comments on ijerph-1259620
Point 1:-17 specimens are a very small sample of fish and being all from the same fisherman (same site) is not the most random sample. I also know that the authors intent to contain the study to the Mar Menor area.
-It is not also clear, that the all sample is 17 organisms or 17 were collected in each season? Please clarify it in the text.
Response 1: The total number of individuals analyzed was 17, distributed as follows: 6 individuals in the June-July 2018 (summer) campaign, 5 individuals in the November 2018-January 2019 (fall-winter) campaign, and 6 individuals in the April-May 2019 (spring) campaign. That question has been included in line 79 as follow:
“Stomach and intestine of 17 individual of commercial gilthead”
Point 2: Please add some information about Fulton’s factor and why it is used?
Response 2: Fulton's condition factor (K) allows to analyze changes in populations subjected to pressures. It is calculated from the weight (g) and total length (cm) of the individuals. According to this formula, when the K factor is equal to or greater than the unit (K=1; K>1) the species is in good health and therefore the environmental conditions are good for its development, when this factor is less than the unit (K<1) then the environmental conditions are unfavorable and are affecting its normal development.
That question has been included in lines 107-108 as follow:
“In order to analyze how the environmental conditions of aquatic ecosystems affect fish populations” Fulton’s factor was also calculated
Point 3: Ln 178 - typing error “Posible” – did the author mean possible?
Response 3: Posible has been changed for “possible” (line 155)
Point 4: Ln 231 – Please add the abbreviation used in the legend of the Table 1.
Response 4: The abbreviation has been added (line 210) as follow:
Table 1. Average values (± standard error) of diferent parameters analysed in commercial gilthead seabream (CGS). (ML: microlitter, MP: microplastics, FB: fibers; MPP: particulate microplastics)
Point 5: Ln 254-261- This sentence needs to be better explained because it is not clear to the reader.
Response 5: The sentence has been rewritten as follow (line 235-239)
However, due to the damage and potential for degradation during the microfiber digestion procedure, characteristics derived from [21] and [22] were also adapted for their identification, including no cellular or organic visible structures, color homogeneity properties and microfibers did not segment or fragmented when pressed.
Point 6: Ln 282 and after in the manuscript when the author mentioned MP quantification include or not ML? I think that need to be clear to the reader otherwise what’s the purpose of differentiating it the M& M section?
Response 7: The quantification of MP does not include ML. However, ML quantification does include MP, since ML is defined as both plastic and non-plastic microparticles. In this study, only that microlitter (ML) whose FTIR analysis confirmed that it was a plastic polymer was counted as microplastics (MP). The purpose is to demonstrate the importance of using an analytical technique that confirms that the detected microliter is plastic and thus avoid overestimating the results. All these issues are explained in lines 224-234.
Point 7: Ln 344 . This sentence needs English spell check.
Response 7: The sentence has been rewritten (325-328) as follow:
Moreover, the results reported in many studies do not clarify whether they refer to the wet or dry weight of the whole sample, or to the wet or dry weight of the tissue or organ containing MP.
Point 8: Ln 397-399. What means being secondary microplastics. The authors did not mention it before not even in Introduction section. That needs to be clarified and its implications of being secondary…
Response 8: A short definition about secondary microplastics has been included (line 384) "Taking into account all these results, we can verify that most of microplastics detected in this study were secondary microplastics, from the continuous fragmentation of oversized plastics."